# β-lactam Resistance in *Pseudomonas aeruginosa*: Current Status, Future Prospects

**DOI:** 10.3390/pathogens10121638

**Published:** 2021-12-18

**Authors:** Karl A. Glen, Iain L. Lamont

**Affiliations:** Department of Biochemistry, University of Otago, P.O. Box 56, Dunedin 9054, New Zealand; gleka251@student.otago.ac.nz

**Keywords:** antibiotic resistance, nosocomial infection, cystic fibrosis, carbapenem, cephalosporin, β-lactamase, carbapenemase, PBP3, AmpC, antibiotic efflux

## Abstract

*Pseudomonas aeruginosa* is a major opportunistic pathogen, causing a wide range of acute and chronic infections. β-lactam antibiotics including penicillins, carbapenems, monobactams, and cephalosporins play a key role in the treatment of *P. aeruginosa* infections. However, a significant number of isolates of these bacteria are resistant to β-lactams, complicating treatment of infections and leading to worse outcomes for patients. In this review, we summarize studies demonstrating the health and economic impacts associated with β-lactam-resistant *P. aeruginosa*. We then describe how β-lactams bind to and inhibit *P. aeruginosa* penicillin-binding proteins that are required for synthesis and remodelling of peptidoglycan. Resistance to β-lactams is multifactorial and can involve changes to a key target protein, penicillin-binding protein 3, that is essential for cell division; reduced uptake or increased efflux of β-lactams; degradation of β-lactam antibiotics by increased expression or altered substrate specificity of an AmpC β-lactamase, or by the acquisition of β-lactamases through horizontal gene transfer; and changes to biofilm formation and metabolism. The current understanding of these mechanisms is discussed. Lastly, important knowledge gaps are identified, and possible strategies for enhancing the effectiveness of β-lactam antibiotics in treating *P. aeruginosa* infections are considered.

## 1. Introduction

*Pseudomonas aeruginosa* is a Gram-negative bacillus that is found in many environments including water and soil, and in association with animals [1]. It is also a major opportunistic pathogen, being one of the most frequent causes of acute infections in hospitalised patients and in patients with predisposing conditions such as severe burns, catheterisation, or neutropenia, causing septicaemia, urinary tract infections, and bacteraemia [2,3,4,5]. *P. aeruginosa* is a primary cause of hospital- and ventilator-acquired pneumonia [6,7,8]. It also causes severe eye infections and chronic infections in patients with cystic fibrosis or chronic obstructive pulmonary disease [9,10,11]. Infections with *P. aeruginosa* are often associated with higher mortality and morbidity than those with other pathogens [12,13].

Examples of *P. aeruginosa* infections that have been intensively studied include chronic infections in people with CF and acute infections in burns patients. Chronic infections in the lungs of people with CF can last for many years and are the leading cause of morbidity and reduced life expectancy in these individuals [14]. The eradication of *P. aeruginosa* infections in individuals with CF is associated with a better long-term outcome [15,16]. *P. aeruginosa* infections in CF are also strongly associated with mortality in childhood [17]. Acute *P. aeruginosa* infections are the leading cause of death in burn victims [18,19]. Treatment of burn wounds becomes increasingly difficult when *P. aeruginosa* infection occurs [20]. In a study of over 5000 patients over twenty years, 55% of burn victim mortality was due to multidrug-resistant *P. aeruginosa* [18]. Other acute *P. aeruginosa* infections, in particular bloodstream infections, also have very high mortality rates [21,22,23,24,25].

*P. aeruginosa* infections can be acquired from most aquatic and damp environments, with these bacteria being commonly isolated from showers, sinks, drains, and even liquid soap [26,27]. For example, in one between 9.7% and 68.1% of tap water samples from intensive care units were contaminated with *P. aeruginosa* [27]. Wash hand basins and other moist areas are a source of hospital outbreaks of *P. aeruginosa* infections [28,29]. Outbreaks have also been linked to a *P. aeruginosa*-contaminated hand soap dispenser [30] and bottled water supplied to an intensive care unit [31]. Analysis of *P. aeruginosa* isolated from newly infected CF patients found that for 13 of 25 patients, the *P. aeruginosa* genotype matched that of isolates obtained from a sink in their house, while one of the isolates matched to the patient’s nebuliser used for antibiotic inhalation, identifying these environments as possible sources of infection [32].

Infections by multidrug-resistant (MDR) isolates (bacteria resistant to one antibiotic from three or more antibiotic classes [33]; Table 1) are particularly problematic. Acute infection with antibiotic-resistant *P. aeruginosa* results in thousands of deaths worldwide each year [5,34,35]. In a meta-analysis of 23 studies of over 10,000 *P. aeruginosa* infections, mortality was 34% in patients with antibiotic-resistant *P. aeruginosa* compared with 22% in those infected with antibiotic-susceptible *P. aeruginosa* [5]. Similar findings resulted from a separate meta-analysis, with hospital patients who had non-MDR *P. aeruginosa* infections having a mortality rate of 24.8% (of 2388 cases) and patients with MDR infections having a mortality rate of 44.6% (of 813 cases) [36]. A factor contributing towards mortality of *P. aeruginosa* infections is the time between the onset of infection and treatment. Treatment within 24 h had a mortality rate of 27.7% compared to a mortality rate of 43.4% if effective treatment was delayed for 24 h [23]. Initial treatment with antibiotics to which the bacteria were resistant was associated with a mortality rate of 40.6%, emphasising the importance of early treatment with antibiotics that are effective against the infecting *P. aeruginosa* [23]. Isolation of infecting bacteria followed by antibiotic susceptibility testing is used to determine which antibiotics are likely to be efficacious. However, the timeframe involved in this process (about two days) can delay effective treatment. For some bacterial species, genomic sequencing and analysis provides a more rapid method to identify effective antibiotics [37] but this approach is not yet available for *P. aeruginosa*.

The frequency of *P. aeruginosa* infections, the clinical challenges and poor outcomes associated with these infections, and in particular the proportion of *P. aeruginosa* isolates that are resistant to antibiotics have resulted in these bacteria being classified as one of a group of six pathogens (the ESKAPE pathogens) that are the most problematic to treat [38,39]. *P. aeruginosa* that have become resistant to the carbapenem class of antibiotics are classified by the World Health Organisation as one of the three “Priority 1: Critical” groups of bacteria for which new treatment strategies are most critically needed [40].

## 2. Antibiotics Used against *P. aeruginosa*

Oral and intravenous delivery of antibiotics can be used to treat a wide range of *P. aeruginosa* infections including septicemia, lung infections, and bone infections, while inhalation of specific antibiotics is also used for the treatment of lung infections in individuals with CF or other forms of lung disease [11,41,42,43]. There are a variety of antibiotic classes used to treat *P. aeruginosa* (Table 1), each having different targets within the bacterial cell.

β-lactams, the topic of this review, inhibit the synthesis of peptidoglycan, a key component of the cell envelope [54]. This inhibits the ability of bacteria to replicate and divide while also reducing the integrity of the cell wall leading to cellular lysis [54]. Aminoglycosides and fluoroquinolones inhibit protein and DNA synthesis, respectively, and polymixins disrupt the bacterial cell membrane. Antibiotics from different classes can be used in combination to increase the likelihood of effectiveness when resistance phenotype is not known and to suppress the emergence of resistance [3].

## 3. The Problem of *P. aeruginosa* Antibiotic Resistance

The low outer membrane permeability of *P. aeruginosa*, coupled to the presence of efflux pumps and other genetic features and its ability to form biofilms, provides it with intrinsic resistance to moderate concentrations of antibiotics including β-lactams [55,56]. Biofilms involve the attachment of cells to a surface by adhesins and encasing the bacterial cells in an extracellular matrix [57]. Biofilms contribute to antibiotic resistance by being difficult for antibiotics to penetrate, because the biofilm lifestyle alters the metabolism of the bacteria, and because biofilms can contain dormant antibiotic-insensitive persister cells [56,57,58,59,60]. Nonetheless, *P. aeruginosa* isolated from the general environment are generally susceptible to antibiotics including β-lactams [61]. However, isolates from clinical settings are frequently resistant to antibiotics, and a high proportion of *P. aeruginosa* infections are by antibiotic-resistant bacteria [62,63]. For example, in one study of hospitalised patients, 37% of 826 *P. aeruginosa* isolates from a CF unit, and 49% of 224 isolates from an intensive care unit were antibiotic-resistant [64]. These isolates were resistant to a wide variety of antibiotics, including gentamicin (58%), carbapenems (55%), and colistin (6%) [65]. In a study of over 1000 *P. aeruginosa* infections in neonatal and paediatric patients, over 10% of isolates were resistant to carbapenems (imipenem, meropenem, doripenem), and approximately 15% to cephalosporins (cefepime and ceftazidime) [66]. Large case studies and meta-analyses have found that people infected with carbapenem-resistant *P. aeruginosa* often [67,68,69] though not always [70], have a significantly higher risk of death than people infected with carbapenem susceptible isolates.

The US Centers for Disease Control and Prevention (CDC) has estimated that there were 32,600 hospitalizations and 2700 deaths from multidrug-resistant *P. aeruginosa* infections in the USA in 2019 [34]. In Europe, there were on average approximately 72,000 infections with antibiotic-resistant *P. aeruginosa* and 4155 attributable deaths per million people in the period 2007–2015 [71], with the numbers of such infections increasing by about three-fold over the time period. The median cost per hospitalization and treatment of patients with antibiotic-resistant *P. aeruginosa* infections was US$99,672, while for patients with susceptible *P. aeruginosa* infections the median cost was US$69,502 [5]. Contributing factors of the increased financial burden of antibiotic-resistant *P. aeruginosa* infections are the increased price of drug regimens and increased days of mechanical ventilation (15 versus 11 days) and increased hospitalization length [5,72]. The total healthcare costs associated with infections by antibiotic-resistant *P. aeruginosa* in the USA were estimated at US$767 million in 2017 [34].

*P. aeruginosa* is capable of developing resistance to all classes of antibiotics through chromosomal mutations [73]. This is well studied in chronic CF infections, where patients are commonly infected with an antibiotic-sensitive *P. aeruginosa* isolate that develops resistance over the course of prolonged infection and treatment [74,75]. Likelihood of infecting *P. aeruginosa* being β-lactam resistant increases with age in CF patients, supporting a model whereby antibiotic resistance develops over the course of treatment [76]. Eradication of *P. aeruginosa* infections becomes increasingly difficult as antibiotic resistance occurs, resulting in worsening patient conditions [15,16].

Severe acute burn wounds can be directly infected with antibiotic-resistant *P. aeruginosa* [77]. Outbreaks of MDR and extensive drug-resistant (XDR; resistant to all but one or two classes of antibiotics) strains are a large issue in burn care units as treatment becomes increasingly difficult [78,79]. A 2016 outbreak of an XDR *P. aeruginosa* infected 10 patients within a burn unit resulting in the death of two patients from septic shock [78]. Although the development of antibiotic resistance is not commonly observed in burn wound infections with sensitive *P. aeruginosa* strains (because of shorter infection times)*,* a case study has shown that a sensitive *P. aeruginosa* strain became resistant within 14 days from the beginning of treatment [13]. Analysis of wastewater from a burns hospital found that out of 100 *P. aeruginosa* isolates 66% were MDR [64], illustrating how antibiotic-resistant isolates can be spread into the general environment.

## 4. Penicillin-Binding Proteins and Peptidoglycan Synthesis

Penicillin-binding proteins (PBPs) are, as their name suggests, the key targets of β-lactam antibiotics. They are involved in the synthesis of the layer of peptidoglycan that forms part of the cell envelope and provides structural integrity to *P. aeruginosa* cells [80]. The peptidoglycan layer consists of linear β(1–4)-Linked disaccharides of N-acetylglucosamine (NAG) and N-acetylmuramic acid (NAM) strands containing on average about 24 NAG-NAM units that are cross-linked together by amide bonds between peptide side chains [81,82]. This structure resists internal cellular pressures while also giving cells their inherent shape [82]. The peptide side chains are synthesised as pentapeptide precursors, with cross-linking between diaminopimelic acid and D-alanine residues bridging different strands of NAG-NAM polymers [83] (Figure 1). Cross-links are formed in a process involving the removal of a terminal D-alanyl residue. In mature peptidoglycan, approximately 50% of the peptide units are cross-linked. In uncross-linked peptides, approximately 40% of all D-alanine have been removed by LD-carboxypeptidases that remove the D-alanine at position four of the peptide sidechain, preventing cross-linking [83].

PBPs are involved in the later stages of peptidoglycan synthesis and remodelling of peptidoglycan during cell growth and division [84,85]. *P. aeruginosa* has eight PBPs, numbered 1a, 1b, 2, 3, 3a, 4, 5, and 7 in order of decreasing molecular mass. The PBPs fall into two categories, high molecular mass (HMM) PBPs (1a, 1b, 2, 3, and 3a) and lower molecular mass (LMM) PBPs (4, 5, and 7) [86]. PBPs play a role in cellular division and controlling cellular morphology through the incorporation of NAG-NAM units into growing peptidoglycan chains via a glycosyltransferase activity, cross-linking of different NAG-NAM units through their peptide side chains, and modifying peptidoglycan peptide chains (Figure 1) [87,88]. There is functional redundancy of PBPs, with only PBP3 being essential for growth [79]. The transpeptidase domains of all PBPs contain three amino acid sequence motifs Ser-XX-Lys (catalytic serine), Ser-X-Gln, and Lys-Ser-Gly-Thr [89], which all play a role in DD-transpeptidase activity (cross-linking of NAG-NAM chains), DD-carboxypeptidase activity (cleaving terminal D-Ala from peptide chains), and DD-endopeptidase activity (cleaving the peptide cross-link made by DD-transpeptidase activity) [87,90]. HMM PBPs 1a and 1b have both glycosyltransferase and transpeptidase domains, while PBPs 2, 3, and 3a only have transpeptidase domains [86,91,92]. Glycosyltransferase domains catalyse the extension of NAG-NAM polymers by incorporation of NAG-NAM units and DD-transpeptidase domains catalyse peptide cross-linking. The DD-transpeptidase domains bind to the pentapeptide precursor and catalyse the formation of a cross-link between the D-alanine residue at the fourth position of a pentapeptide side chain and a diaminopimelate residue of a tri-, tetra-, or penta-peptide of an adjacent peptide sidechain, on a different NAG-NAM strand (Figure 1) [84]. During the formation of this cross-link, the terminal D-Ala of the pentapeptide is removed [84].

LMM PBPs in *P. aeruginosa* have DD-carboxypeptidase activity [93,94]. Deletions of genes encoding LMM PBPs increased the presence of pentapeptides in peptidoglycan which indicates the lack of DD-carboxypeptidase activity [93]. PBP5 is the main DD-carboxypeptidase, followed by PBP4 and, lastly, PBP7 [93]. The LMM PBP PBP4 also regulates *ampC* (β-lactamase) expression [95]. Although deletion of all three LMM PBPs (4, 5, and 7) cause a significant increase in pentapeptide levels, there was no significant effect on cell morphology [93].

As well as being required for peptidoglycan synthesis, PBPs play a role in peptidoglycan recycling. Recycling occurs as part of the process of peptidoglycan turnover and remodelling that is required for cell growth and division. Recycling of peptidoglycan components released during hydrolysis by lytic transglycosylases and carboxypeptidases reduces the metabolic burden of peptidoglycan synthesis [96], as described below. LMM PBPs with endopeptidase activity form an important part of this process [96]. PBP4 has both a DD-endopeptidase and DD-carboxypeptidase activity [97,98].

## 5. β-lactam Antibiotics: Inhibitors of PBPs

Antibiotics in the β-lactam class have bactericidal activity against a broad spectrum of bacteria [99]. The bactericidal properties of β-lactams are due to their ability to inhibit the transpeptidase and DD-carboxypeptidase activities of PBPs [54]. β-lactams are structural mimics of the terminal D-alanine-D-alanyl residues of peptidoglycan pentapeptide precursors which PBPs bind to perform their catalytic functions (Figure 2) [100,101]. Once the β-lactam has entered the active site of a PBP, it covalently binds to the catalytic serine in the Ser-XX-Lys motif permanently, inactivating the PBP [102,103]. The inhibition of PBPs can result in filamentation and can also reduce the structural integrity of the cell wall resulting in lysis of the bacteria [104,105,106].

β-lactam antibiotics all have a core structure of a four-membered lactam ring which is closed by an amide bond (Figure 2) [108]. β-lactams are assigned into subclasses on the basis of the nature of the chemical groups attached to this core structure, with subclasses containing multiple members [100]. The differences in side chains affect many characteristics of the β-lactams, including the affinity towards different penicillin-binding proteins PBPs, ability to cross cell envelopes, their chemical stability, and resistance to degradation by β-lactamases [100].

The affinities of anti-Pseudomonal β-lactams for different PBPs are shown in Table 2. Several β-lactams have a high affinity for a number of different PBPs. Notably, however, all of those that have a low minimum inhibitory concentration (MIC) and are used in clinical practice have a high affinity for PBP3, the only essential PBP in *P. aeruginosa*. Conversely, cephalexin that is not clinically effective has a low affinity towards PBP3 and is not effective at killing *P. aeruginosa* (high MIC). Faropenem has a high affinity for PBP3 but nonetheless is inefficient at killing *P. aeruginosa* demonstrating that affinity for PBP3 is not by itself sufficient to ensure effectiveness. This high MIC for faropenem is attributed to a combination of intrinsic resistance mechanisms [109].

As well as causing cell lysis and filamentation, in *E. coli,* inhibition of PBPs can lead to unnecessary recycling of the cell wall, depleting cellular resources and contributing to cellular death [104]. This process depends on the product of the *slt* gene, a transglycosylase [104]. The *slt* gene product of *P. aeruginosa* appears to have an equivalent function to its *E. coli* homologue [113,114,115] and may contribute to cellular resource depletion in *P. aeruginosa* in the presence of β-lactams.

The mode of antibiotic delivery and the use of combinations of different antibiotics have been found to be important in the treatment of *P. aeruginosa* infections. Continuous infusion of meropenem, aztreonam, and ceftazidime can eradicate MDR *P. aeruginosa* infections [116,117]. This finding is supported by a meta-study which found that continuous infusion of meropenem had a higher success rate than intermittent dosages for MDR *P. aeruginosa* [118]. Recent studies have indicated that dual antibiotic regimens improve the survival of patients with MDR *P. aeruginosa*. The combination of a carbapenem antibiotic with colistin increased the eradication of *P. aeruginosa* infections [42]. Dual β-lactam therapy also has been shown to have enhanced efficacy in killing *P. aeruginosa* [119], with an example being ceftazidime-avibactam paired with meropenem [120].

## 6. Mechanisms of β-lactam Resistance

Antibiotic-resistant isolates of *P. aeruginosa* arise through genetic changes in antibiotic-susceptible bacteria (acquired resistance). Acquired resistance can occur through mutations affecting a wide range of cellular functions. The primary mechanisms for the development of β-lactam resistance through mutation include alterations to the PBP3 target protein, decreased antibiotic uptake, increased export, and degradation of antibiotic molecules (Figure 3) [121]. In addition, horizontal gene transfer can lead to the acquisition of antibiotic-degrading enzymes (β-lactamases) from other bacteria [122]. Metabolic changes and increased biofilm production may also play a role in resistance [123]. Resistance mechanisms are described in detail below.

β-lactam resistance can arise from genetic changes that reduce antibiotic uptake through porins, increase degradation of β-lactams, alter the PBP3 target protein, or increase antibiotic efflux. Resistance often involves a combination of these mechanisms. CM, cytoplasmic membrane; P, periplasm; OM, outer membrane.

The availability of methods for relatively inexpensive whole-genome sequencing has greatly accelerated the discovery of genetic changes that contribute to resistance. A number of studies have used experimental evolution to develop β-lactam-resistant *P. aeruginosa* from sensitive strains, followed by whole-genome sequencing to identify the mutations causing resistance [124,125,126,127]. Whole-genome sequencing of isolates from chronically-infected patients has shown that genes acquiring mutations during resistance development in vitro also acquire mutations that are likely to contribute to resistance during infection [128]. Whole-genome sequencing of clinical isolates has also enhanced understanding of the contributions of horizontally acquired genes to β-lactam resistance. Collectively, these studies allow a good understanding of the mechanisms of β-lactam resistance in *P. aeruginosa*. Different resistance mechanisms are discussed in detail below.

## 7. Target-Site Modification: Changes to PBP3

Mutational changes to PBPs that are associated with resistance act by reducing the affinities of PBPs for β-lactams [129,130,131,132]. PBP3 is encoded by the *ftsI* gene. The essential nature of PBP3 in *P. aeruginosa* was shown through the conditional expression of the *ftsI* gene using an inducible promoter [86]. Inhibition of *ftsI* expression led to long filaments of cells, indicating a defect in cell division. As the only essential PBP in *P. aeruginosa*, PBP3 is the primary target for β-lactams [86]. Experimentally-evolved mutants of *P. aeruginosa* resistant to β-lactams frequently have mutations in *ftsI* [124,126,132], and PBP3 sequence variants are also common in isolates from patients that have reduced β-lactam susceptibility [133,134,135].

The structure of PBP3 bound to different β-lactams has been determined [102]. PBP3 is comprised of a short cytoplasmic N-terminal domain, a transmembrane helix, a domain predicted to play a role in protein-protein interactions, and a transpeptidase domain [136,137] (Figure 4). The active site of PBP3 contains the protein sequence motifs Ser-XX-Lys (residues 294–297), Ser-x-Asn (residues 349–351), and Lys-Ser-Gly-Thr (residues 484–487) that are present in all PBPs [136,137]. β-lactams bind to the catalytic serine (S294) of the Ser-XX-Lys motif [102]. Ceftazidime, aztreonam, meropenem, and imipenem all bind at the active site but cause different conformational changes [102]. β-lactams that are effective anti-Pseudomonas agents, including aztreonam, meropenem, imipenem, doripenem, ceftazidime, and ceftolozane, all have a high affinity for PBP3 [42,43] (Table 2) emphasising that PBP3 is a key β-lactam target.

Variants of PBP3 likely to contribute to β-lactam resistance in clinical isolates were identified by comparing the PBP3 sequence of antibiotic-resistant clinical isolates with those of antibiotic naive strains (environmental or pre-treatment) [126,128,135,138,139,140,141,142,143,144,145,146,147]. The most frequent PBP3 variants that are likely to affect β-lactam activity are amino acid substitutions that, as might be expected, are centred around the active side and, in particular, are near the catalytic serine of the Ser-XX-Lys motif (Figure 4, Appendix A). Variants in experimentally-evolved resistant bacteria also cluster around the active site and are often at the same residues as those in clinical isolates [124,125,126,127,148]. Variants around the active sites of PBPs also contribute to β-lactam resistance in other species [131,149,150], and some cases have been shown to reduce the affinity of β-lactams for the PBP [150,151]. It seems likely that sequence variants around the active sites in PBPs cause slight confirmational changes that reduce the ability of β-lactams to bind and the active site and react with the catalytic serine [129,132]. Consistent with this prediction, the PBP3 variant F533L reduces the affinity of PBP3 for meropenem [132]. It is currently not known whether, or how, PBP3 variants away from the catalytic domain, such as the frequently observed G63C/D variants (Figure 4, Appendix A), contribute to β-lactam resistance.

As different β-lactams bind slightly differently to the PBP3 active site, mutations around the active site are likely to have different effects with different β-lactams. There is some evidence to support this prediction. Analysis of isolates from a CF patent with PBP3 variants V465G or A244T showed that both variants were associated with aztreonam and cefsulodin resistance, whereas only the V465G variant was associated with ceftazidime and piperacillin resistance [140]. In a separate study, the PBP3 variant R504C was associated with ceftazidime and cefsulodin resistance, whereas the variant P527S was associated with resistance to aztreonam, cefepime, ceftazidime, and cefsulodin [134]. Biochemical assays are limited, but in one study the sequence variant A539T reduced affinity for meropenem and ceftazidime, and sequence variant F533L selectively reduced affinity for meropenem but not ceftazidime [132]. Collectively these findings support the notion that different PBP3 sequence variants can have different impacts on different antibiotics, but more work is needed to fully understand the relationships between PBP3 sequence variants and β-lactam resistance.

## 8. Reduced Uptake of β-lactams: The Role of Porins

Variants in the *oprD* gene are commonly found in isolates of *P. aeruginosa* that are resistant to carbapenems [76,152,153,154,155,156,157,158]. The *oprD* gene encodes the OprD porin that plays a role in the uptake of small basic amino acids and small peptides. OprD also mediates uptake of carbapenems other than faropenem [153,159]. Basic amino acids act as competitive inhibitors towards the uptake of carbapenems, and so the growth environment influences carbapenem susceptibility [160]. Disruption of the *oprD* gene via point mutations, frameshifts, premature stop codons, or large deletions all contribute to carbapenem resistance [153,155,156,157]. These mutations reduce or abolish the uptake of carbapenems through OprD [152,157,158]. In addition to contributing to carbapenem resistance, loss of *oprD* increases the ability of *P. aeruginosa* to colonize mucosal environments, and increases resistance towards acidic environments in a mouse model of infection [161].

Mutations to OprD do not affect susceptibility to other classes of β-lactams, indicating that this porin is not involved in their uptake [159,162,163]. How antibiotics in these classes access the periplasm of *P. aeruginosa* is not fully understood [162]. Most porins (other than OprD) do not mediate antibiotic uptake [162]. However, loss of porin OpdP confers a reduced susceptibility to meropenem and, in an OprD-lacking strain, to imipenem and doripenem [163]. Heterologous expression assays confirmed that OpdD can contribute to susceptibility to carbapenems presumably by providing a channel into the periplasm [163]. Loss of OprF may cause a slight increase in piperacillin resistance [162], and loss of OpdH reduced susceptibility to ceftazidime but not to other cephalosporins [164], suggesting that these porins may also play a role in the entry of β-lactams into *P. aeruginosa*. Diffusion through the lipid layer of the outer membrane likely also plays a role in the uptake of β-lactams [165].

## 9. *P. aeruginosa* Efflux Systems

Most isolates of *P. aeruginosa* contain 12 Resistance-Nodulation-Division efflux system pumps, which play a role in virulence, stress response, and both intrinsic and acquired antibiotic resistance [109,166,167,168]. Each of these efflux pumps is comprised of a protein that spans the cytoplasmic membrane, an outer membrane protein, and a periplasmic component that links the two [169]. Efflux pumps form channels from the cytoplasm to the outside of the cell that export a wide range of substrates, in a process driven by proton motive force [169]. As well as compounds in the cytoplasm, chemicals present in the periplasm can be exported. Different pumps export different compounds, although how substrate selection occurs is not well understood. Overexpression studies have shown that efflux pumps MexAB-OprM, MexXY-OprM, and MexCD-OprJ are the most important in the context of β-lactam resistance [170,171,172,173,174], although each pump can export a wide range of antibiotics (Table 3). These efflux systems are clinically relevant as they are overexpressed in many antibiotic-resistant clinical isolates [76,138,152,156,166,175,176,177,178,179]. For example, in one multicentre study, 39 of 80 CF isolates overproduced at least one of these efflux systems, with 65 having increased expression of MexXY-OprM, 36 of MexAB-OprM, and two of MexCD-OprJ [76].

The MexAB-OprM efflux pump plays a role in the intrinsic resistance of *P. aeruginosa* to a wide range of β-lactams, with deletion of the *mexABoprM* operon increasing susceptibility to faropenem, sulopenem, ritipenem, temocillin, and ticarcillin [109,167]. Increased *mexABoprM* expression (acquired resistance) is often observed in carbapenem-resistant clinical isolates–for example, 16 of 23 isolates and 28 of 32 isolates in two separate studies [152,175]–and contributes to resistance to a wide range of β-lactams including meropenem, ceftazidime, aztreonam, ticarcillin and carbenicillin [185,186].

Overexpression of the *mexABoprM* genes occurs because of mutations affecting the repressor proteins MexR, NalC, or NalD [179]. Deletion of the *mexR* gene leads to the highest levels of *mexABoprM* expression, followed by deletion of *nalD* and *nalC* [186,187]. MexR, the primary regulator of the *mexABoprM* operon, plays a role in sensing oxidative stress. MexR binds to the promoter of the *mexAoprM* operon repressing expression, but under conditions of oxidative stress, MexR disassociates increasing expression of the efflux pump [188,189]. Mutations in the DNA-binding domain of MexR inhibit its ability to bind to the *mexABoprM* promoter, increasing expression [190]. Overexpression of *mexR* significantly increases susceptibility to aztreonam consistent with its role in the repression of *mexABoprM* and the involvement of this efflux pump in aztreonam resistance [190]. NalD is a second repressor of the *mexABoprM* operon, acting similarly to *mexR*, and mutations in *nalD* cause increased expression of the *mexABoprM* genes and are associated with β-lactam resistance [186,187]. Mutations in *nalC* lead to overexpression of gene PA3719, which in turn leads to *mexABoprM* overexpression [191].

Increased *mexXYoprM* expression also contributes to resistance to multiple β-lactams (Table 3). Expression of the *mexXY* operon is regulated by the repressor MexZ that binds to the promoter of the *mexXY* operon [192]. Disruption of protein synthesis causes increased synthesis of the AmrZ protein that interacts with MexZ, dislodging it from the promoter of the *mexXY* operon and inducing expression [171]. MexXY uses OprM from the *mexABoprM* as the outer membrane component [193]. Mutations in *mexZ* are the most common cause of *mexXY* overexpression in clinical isolates [194,195].

MexCDOprJ is capable of exporting a wide variety of antibiotics (Table 3). The *mexCDoprJ* operon is regulated by the repressor NfxB [73,196], and expression is induced by membrane-damaging agents [197]. Mutations in *nfxB* can cause *mexCDoprJ* overexpression, although this is infrequently observed in clinical isolates [76,198,199]. Mutations in *mexD* can lead to increased resistance to cephalosporin-β-lactamase inhibitor combinations (ceftolozane-tazobactam and ceftazidime-avibactam) through altered substrate specificity of MexCDOprJ [200].

Increased expression of the efflux pump *mexEF-oprN* is associated with imipenem resistance [201], although MexEF-OprN does not export β-lactams [202]. Overexpression of the *mexEF-oprN* genes occurs because of mutations affecting the repressor protein NfxC and MexS that influence the expression of the *mexABoprM* operon and the *oprD* gene [76,183,184,203]. Increased expression of *mexEF-OprN* is, therefore, likely to be a consequence of mutations that affect antibiotic susceptibility, rather than a direct contributor to β-lactam resistance.

It should be noted that overexpression of efflux pumps can also increase antibiotic susceptibility, with overexpression of both MexEF-OprN and MexCD-OprJ making *P. aeruginosa* more susceptible to imipenem, ticarcillin, aztreonam, and aminoglycosides [172,204]. Therefore, the benefits of overexpressing efflux pumps are likely dependent on the environment, with different antibiotics selecting for or against the expression of different efflux pumps.

## 10. Degradation of β-lactams by β-lactamases

β-lactamases are enzymes that cleave open the β-lactam ring of β-lactam antibiotics through hydrolysis, inactivating the antibiotic [205]. They are categorised into four classes (A to D) based on their amino-acid sequence similarity [205,206]. Within each class, enzymes are further categorised into families based on the protein sequence. Families are named on the basis of substrate β-lactam, or geographic location where they were first identified. Enzymes in classes A, C, and D have a catalytic serine for substrate hydrolysis. Class B enzymes are metallo-β-lactamases that catalyse the hydrolysis of β-lactam rings in reaction mechanisms involving a metal ion, most commonly a zinc ion [206,207,208]. Due to their different mechanism of action metallo-β-lactamases have likely had different evolutionary origins from the other classes of β-lactamases [207]. Metallo-β-lactamases have a broad activity spectrum degrading all β-lactams except monobactams [209]. The presence of Metallo-β-lactamases is significantly associated with carbapenem resistance [210]. β-lactamases that are capable of degrading carbapenems (carbapenemases) are especially problematic because of the critical role of carbapenems in managing *P. aeruginosa* infections [211]. Carbapenemase-producing isolates are often a cause of severe infections and are becoming more frequently detected in hospitals [67,68,69,212,213,214,215,216]. Almost all carbapenemases are class A, B, or D β-lactamases as class C β-lactamases have only low activity against carbapenems. For a detailed review of carbapenemases, see [217].

The effectiveness of β-lactamases is reduced by enzyme inhibitors and these are often co-administered with β-lactam antibiotics, increasing antibiotic efficacy. β-lactamase inhibitors used in treating *P. aeruginosa* infections include clavulanate, tazobactam, avibactam, and vaborbactam for Class A β-lactamases, relebactam for Class A and Class C enzymes, and avibactam for class C and a limited number of class D β-lactamases [217,218,219,220,221]. Class B enzymes can be inhibited by metal ion chelators, but currently, these are not in clinical use as β-lactamase inhibitors [217,222].

## 11. β-lactamases Encoded by the Core Genome

Like many other Gram-negative bacteria, *P. aeruginosa* has a chromosomally encoded Class C β-lactamase, AmpC [73,205]. Class C β-lactamases have high activity against penicillins and cephalosporins [206,223]. AmpC contributes to intrinsic resistance to many penicillins, including faropenem, ritipenem, and sulopenem, as shown by increased susceptibility to these antibiotics when the *ampC* gene is deleted [109]. Antibiotic-resistant *P. aeruginosa* clinical isolates often have high levels of *ampC* expression, reducing susceptibility to ceftazidime, cefepime, aztreonam, and piperacillin, although having little or no effect on susceptibility to carbapenems [224,225]. Expression of the *ampC* gene is regulated through a complex signalling pathway. Increased expression of *ampC* can occur through activation of this pathway by the presence of β-lactams (Figure 5), or by mutations that alter the pathway.

The expression of *ampC* is controlled by the transcriptional regulator AmpR (Figure 5A) [226]. During peptidoglycan synthesis and recycling, the peptidoglycan precursor UDP-NAM pentapeptide is formed and binds to AmpR. The D-Ala-D-Ala of the pentapeptide plays a primary role in interacting with AmpR [227]. The resulting complex binds to the divergent *ampC*-*ampR* promoter and inhibits transcription of *ampC* [95,228]. In the presence of β-lactams, there are increased amounts of NAG-NAM pentapeptide units formed following hydrolysis of mature peptidoglycan (Figure 5B). These are imported into the cytoplasm through the AmpG permease and they, as well as NAM penta- and tri-peptides generated by removal of the NAG moiety, bind to AmpR. The binding of these molecules causes AmpR to activate the expression of *ampC* [95,228].

Peptidoglycan fragments that have been imported into the cytoplasm are processed for recycling by NagZ, AmpD, and other enzymes [95,226,228,229,230] (Figure 5). Inhibition of PBPs increases the intracellular concentrations of NAG-NAM pentapeptide, NAM pentapeptide, and NAM tripeptide [95]. Excess pentapeptides arising from the action of β-lactams are thought to saturate AmpD, raising the intracellular concentration of the AmpR-activating compounds [95,228]. The cytoplasmic concentration of NAG-NAM pentapeptide is also influenced by LMM PBPs, in particular the PBP4 protein [95]. Inhibition of PBP4 by mutation or β-lactams is a major inducer of *ampC* expression [93]. Inhibition of PBP4 does not significantly increase NAG-NAM pentapeptide levels in the periplasm [83]. Instead, inhibition of PBP4 is thought to increase peptidoglycan recycling [95,97] resulting in an increase in the intracellular concentration of NAG-NAM pentapeptide.

Mutations in *ampR* contribute to β-lactam resistance [231,232]. These mutations inhibit the UDP-NAM pentapeptide from binding to AmpR leading to constitutive expression of *ampC* at high levels [97,233,234]. Mutations in *dacB* that encodes PBP4 are also found commonly in β-lactam resistant clinical isolates and lead to increased *ampC* expression [178,235]. PBP4 inhibition also activates the CreBC signalling [236], which is a global regulator of bacterial fitness, biofilm development, and *ampC* expression. It is not yet known how PBP4 inhibition activates CreBC signalling or how CreBC signalling regulates *ampC* expression [236].

Mutations in *ampD* are also often found in clinical isolates, increasing β-lactam resistance [97,230,234,237,238]. AmpD plays a role in recycling peptidoglycan by cleaving NAG-NAM from the peptide side chains that are imported from the periplasm into the cytoplasm [229,230]. Disabling or impairing the function of AmpD through mutation results in increased quantities of partially recycled peptidoglycan building up especially the NAG-NAM tripeptide [95].

As well as reduced β-lactam susceptibility through increased expression of the *ampC* gene, the catalytic activity of AmpC towards many penicillins and most cephalosporins can be increased by mutations altering the enzyme (Figure 6) [238,239,240]. Additionally, *ampC* mutations can reduce the affinity of inhibitors such as avibactam and tazobactam for AmpC [239,240,241,242]. For a full review on *ampC* point mutations see [240].

Two other β-lactamases are encoded in the core genome of *P. aeruginosa.* One is a class A β-lactamase PIB-1 (PA5542) capable of degrading imipenem [243], and the second is a class D OXA-50 like β-lactamase (PA5514/*poxB*) capable of degrading carbapenems [244,245]. Inactivation of PIB-1 increases the susceptibility of *P. aeruginosa* to carbapenems, and overexpression of PoxB reduces susceptibility to meropenem [243,244,245]. The contributions of these enzymes to antibiotic resistance in clinical settings, if any, has not yet been studied.

The structure of AmpC is shown with avibactam (blue) bound to the catalytic serine (pink) at the active site. Deletions that contribute to β-lactam resistance [240] are shown in black. The locations of amino acid variants that contribute to β-lactam resistance [240] are shown in red with the side chains displayed. Sequence variants P180L and F147L reduce susceptibility to ceftazidime and ceftolozane-tazobactam. Variants V239A, G242R, E247K, E247G, and Y249H reduce susceptibility to ticarcillin, ceftazidime, ceftolozane-tazobactam, piperacillin-tazobactam and cefepime, and except for E247K aztreonam. Variants L320P, N373I, and ΔT316-ΔQ321reduce susceptibility to a wide range of cephalosporins. Variants that reduce the effectiveness of the AmpC inhibitor tazobactam are G183V, E247K (shown in red), and the deletion ΔG229–ΔE247 [242]. Variant N347Y (cyan) reduces the effectiveness of the inhibitor avibactam [241]. Amino acid residues are numbered relative to the start codon. The image is based on the crystal structure 4HEF_1 [246].

## 12. β-lactamases Acquired by Horizontal Gene Transfer

Many clinical isolates of *P. aeruginosa* have additional β-lactamases that have been acquired by horizontal gene transfer [209,210,247,248,249]. Enzymes in classes, A, B, and D can all be acquired in this way. The most prevalent horizontally acquired class A β-lactamases in *P. aeruginosa* include enzymes in the SHV, TEM, KPC, and GES families [250,251]. For example, a study in 88 isolates of *P. aeruginosa* from patients in Germany found that 44% had a SHV, 23% had a TEM, 14% had a KPC, and 2% had a GES [250]. A novel class A carbapenemase, GPC-1, has also been recently discovered in *P. aeruginosa* [252].

Amongst class B Metallo-β-lactamases, which include carbapenemases, two of the most prevalent horizontally acquired enzymes are VIM and IMP [210,250,253,254]. Of 207 isolates of *P. aeruginosa* from patients in China, 55% had a class B β-lactamase, of which 32% were a VIM and 29% an IMP with the remainder being in the SIM, NDM SPM, and GIM families [255]. In the previously mentioned study of isolates from German patients, IMP had a prevalence of 16%, and VIM was present in 6% of isolates [250]. The difference in prevalence between the two studies may be due to factors such as a differing distribution of β-lactamases around the globe and different treatment protocols.

Transferable Class C β-lactamases are relatively rare in species such as *P. aeruginosa* that have chromosomally encoded *ampC* [256,257]. Transferable Class C β-lactamases can have activity against penicillins, cephalosporins, and monobactams [256]. These enzymes are thought to have originated from chromosomally-encoded enzymes that have been transferred to mobile elements [257]. Class C β-lactamases present in *P. aeruginosa* as a result of horizontal gene transfer include FOX-4 in a high-risk strain of *P. aeruginosa* (ST308) [258] and CMY-2 that was found in a variety of clinical isolates [259].

Class D β-lactamases that are increasingly problematic in *P. aeruginosa* include the OXA family of β-lactams, named for their high activity against oxacillin [209,249]. Class D β-lactamases have a broad activity spectrum capable of degrading all β-lactams [260]. Many Class D OXA β-lactamases, such as the OXA-10-like β-lactamase that confers resistance to ceftazidime, were first discovered in *P. aeruginosa* [249]. In one study, of 1173 *P. aeruginosa* isolates from patients, 15.4% had OXA β-lactamases [261]. In a separate study, of 75 β-lactamase-producing isolates, OXA-1 was the most common β-lactamase being found in 37.3% of isolates, followed by OXA-4 (in 32%), GES-1 (in 16%), and VEB-1 (in 13.3%) [262]. OXA-1 and OXA-4 were both present in 18.7% of isolates. In a further study of 184 carbapenem-resistant *P. aeruginosa* isolates, the OXA-type carbapenemases present were OXA-23 in 6.5% of isolates, OXA-40 in 0.5% and OXA-58 in 0.5% [263].

The acquisition of carbapenemases by *P. aeruginosa* has contributed towards a significant proportion of overall carbapenemase resistance [264]. In a study of 232 carbapenem-resistant *P. aeruginosa* isolates, 71 isolates had carbapenemases that had likely been acquired through HGT [265]. In *P. aeruginosa* acquisition of β-lactamase genes by horizontal transfer occurs via plasmids or through integrative and conjugative elements (ICEs) that integrate into the chromosome of recipient cells following transfer. For example, the gene encoding an OXA-198 enzyme in isolates of *P. aeruginosa* from Belgium was carried on an IncP-type plasmid [266]. IMP was the first transferable Metallo-β-lactamase found in *P. aeruginosa* (in 1991) and was on a conjugative plasmid [247]. In a separate study, 11 VIM β-lactamases and one IMP β-lactamase from *P. aeruginosa* isolates were found on self-mobilizing plasmids [248]. Conversely, the gene encoding a class A GES-6 enzyme was present on an ICE element in a *P. aeruginosa* [267], and bioinformatic analysis shows that β-lactamases are commonly located on ICEs in *P. aeruginosa* [268]. Multiple antibiotic-modifying genes can be present on a single mobile genetic element [269,270,271]. β-lactamases are often present on integrons that facilitate their capture by mobile genetic elements [270,271,272]. This can result in the presence of multiple β-lactamase genes, such as carbapenemase- and cephalosporinase-encoding genes, on a single mobile element conferring resistance to a wide range of β-lactams [261,269,270,271]. Compounding the problem, integrons can also carry genes conferring resistance to other antibiotic classes such as aminoglycosides, so that horizontal gene transfer can result in multidrug-resistant bacteria [273,274].

Horizontal gene transfer can occur between as well as within species, providing the potential for *P. aeruginosa* to acquire β-lactamases from unrelated bacteria such as the Enterobacteriaceae. KPCs first discovered in *Klebsiella pneumoniae* are now found in *P. aeruginosa* and many other gram-negative bacteria such as *Enterobacter* spp., *E. coli, Proteus mirabilis*, and *Salmonella* spp. [122]. Conversely, VIMs were first discovered in *P. aeruginosa* and are now found widely spread amongst gram-negative bacteria [122].

Carbapenemase-producing bacteria have been found in hospital drainage systems (wastewater) and sewage systems and in the general wastewater downstream of hospital treatment systems [216,275,276,277,278]. This indicates that hospitals may be a large source of the dissemination of carbapenemase-producing isolates into the general environment, providing potential for inter-species gene transfer. However, carbapenemases have also been found in many other environments. Carbapenemase-producing bacteria have been found in most aquatic environments including rivers, the sea, well water, sewage, and drinking water [279,280]. Non-aquatic environments also harbour carbapenemases. In one study, 4 out of 856 bacterial isolates from samples of vegetables had carbapenemases-encoding genes [281]. The widespread presence of carbapenemase-encoding genes in the general environment increases the risk of acquisition of carbapenemase resistance genes by *P. aeruginosa*.

## 13. Lifestyle and Metabolism: Other Contributors to Resistance

The primary mechanisms of β-lactam resistance are outlined above, but mutational changes affecting other pathways can also influence susceptibility to β-lactams. Bacteria near the centre of a biofilm have low metabolism and growth because of low oxygen and nutrient levels, factors that contribute towards β-lactam resistance as β-lactams only kill actively growing cells [57]. Mutations in genes that regulate biofilm production are observed in many clinical isolates of *P. aeruginosa* from the lungs of patients with CF [282,283,284] and may reduce the susceptibility of the bacteria to β-lactams. For example, mutations in *wspF* were present in 68% of clinical isolates that have increased production of extracellular matrix [285]. *wspF* belongs to the Wsp signalling complex, a major regulator of biofilm formation [286]. The loss of *wspF* via mutations is predicted to leave the Wsp signalling complex in an active conformation leading to a signalling cascade upregulating genes responsible for adhesion and biofilm formation [286]. Similarly, point mutation and deletions of *rpoS* increase adhesion and biofilm production [287,288]. In experimental evolution studies, *rpoS* was commonly mutated in *P. aeruginosa* grown as biofilms and then exposed to imipenem [287]. Mutations in *rpoS* have also been found in many clinical isolates [287].

Experimental evolution studies have shown that mutations in a wide variety of other genes are also associated with an increased ability to tolerate β-lactams [55]. Many of these mutations affect lipopolysaccharide (LPS) synthesis (e.g., *wapR/galU*) [289] or alter other aspects of metabolism (e.g., *aroB/acyl-CoA thiolase*) [124,290] or membrane composition. Mutations in the *galU* gene that plays a role in the synthesis of lipopolysaccharide occur in CF clinical isolates of *P. aeruginosa* [135] and increase ceftazidime and meropenem tolerance [74]. How these mutations contribute towards β-lactam resistance is not well studied. It may be that altered LPS synthesis reduces the permeability of the outer membrane for β-lactams. There is some evidence that mutations affecting LPS synthesis can reduce bacterial fitness in vitro [291] but whether they affect fitness in vivo is not known.

Mutations that lead to enhanced activity of AlgU, a regulator of alginate biosynthesis, are predicted to cause a metabolic burden, and mutations in *rpoN* which nitrogen metabolism, reducing the growth rate in many clinical isolates [292]. Slower growth rates in *P. aeruginosa* are associated with antibiotic resistance [292]. Mutations in genes encoding enzymes of metabolism, such as triosephospate isomerase (central carbon metabolism), N-acetylglutamate synthase (arginine metabolism), and 3-dehydroquinate synthase (synthesis of aromatic amino acids) also increase tolerance to β-lactams [55,124,289]. Whether these mutations act by reducing growth rate by altering antibiotic susceptibility because of reduced cellular respiration [293] is not yet clear.

During chronic infection, *P. aeruginosa* can also evolve morphotypes termed small colony variants (SCVs) because of their appearance during growth on agar plates [294]. SCVs have an increased ability to resist antibiotics. SCVs arise from changes to signalling pathways, including the Wsp signalling pathway described above, and also to changes in metabolic pathways [295]. However, the relationship between metabolic changes, the SCV phenotype, and the increased ability to tolerate β-lactams is not yet fully understood.

## 14. Conclusions and Prospects for the Future

β-lactam antibiotics are a key tool in the treatment of *P. aeruginosa* infections and will remain so for the foreseeable future. However, the occurrence of resistant isolates of this major and problematic pathogen, complicating treatment, is a major concern. Clinical isolates that have acquired carbapenemases are a serious problem as carbapenems are one of the last lines of defence against *P. aeruginosa* [211]. The importance of β-lactams in treating *P. aeruginosa* infections, and the widespread occurrence of antibiotic-resistant bacteria, has led to a large amount of research into the mechanism of β-lactam action and bacterial resistance. Consequently, we have a good understanding of how β-lactams act and on resistance mechanisms, including mutations that reduce carbapenem uptake or upregulate efflux pumps, target site mutations that alter PBP3 or increase AmpC expression and activity, and acquisition of antibiotic modifying enzymes through HGT.

Nonetheless, there are still some important knowledge gaps. Sequence variants in PBP3 are an important contributor to resistance, but the effects of different variants on affinity for, and effectiveness of, different antibiotics are not yet fully understood. A better understanding of the effects of PBP3 variants on β-lactam affinity will be important in the design of new β-lactams, as well as helping to understand cross-resistance between currently used antibiotics. Some other resistance mechanisms are class-specific. For example, mutations altering OprD contribute to carbapenem resistance but not cephalosporin resistance, whereas mutations affecting PBP4 increase resistance to cephalosporins but not carbapenems. β-lactamases acquired by horizontal gene transfer also exhibit specificity for different classes of β-lactams, as well as different susceptibilities to β-lactam inhibitors. Refining our understanding of class-specific resistance mechanisms has the potential to allow fine-tuning of β-lactam use in clinical practice.

More broadly, while the primary contributors to β-lactam resistance are relatively well understood, the relationship between genotype (genome sequence) and phenotype is not yet fully clear. How do different combinations of resistance mechanisms affect β-lactam susceptibility, what is the contribution of intrinsic resistance and to what extent does it vary between isolates? It is especially difficult to determine if variants in gene promoters and regulatory regions or non-coding RNAs have an effect. Fully understanding the relationship between genotype and phenotype has the potential to allow the development of genome-based tools for rapid prediction of antibiotic susceptibility [296,297], as has been done for other bacterial species [37,298,299,300]. More rapid treatment with antibiotics would reduce the mortality associated with *P. aeruginosa* infections.

How can the occurrence of *P. aeruginosa* resistant to β-lactams be minimised? One approach that has already been implemented in a variety of settings is antimicrobial stewardship–limiting β-lactam use to cases where there will be a clear benefit [301]. Initial studies applying this approach to *P. aeruginosa* are encouraging, with the rate of antibiotic resistance (including β-lactam resistance) being lowered once stewardship programmes were implemented [302,303].

Additional approaches to minimise the emergence of resistant bacteria are to use combinations of antibiotics or to avoid prolonged exposure to an antibiotic in chronic infections by alternating between different antibiotic classes–“antibiotic cycling” [304]. The first of these relies on the principle that if an infecting bacterium has become resistant to one antibiotic, it will still be susceptible to a second. As well as antibiotic combinations, phage therapy in parallel with antibiotic treatment is undergoing extensive testing in clinical trials and results are encouraging [305,306,307,308,309]. Powdered phage cocktails are being specifically designed for the treatment of respiratory infections [310]. Peptides with anti-microbial activity are also being actively explored as potential tools in overcoming antibiotic-resistant bacteria [311] and peptide–β-lactam combinations may present another approach to preventing the emergence of resistance.

The “antibiotic cycling” approach is based in part on the concept that mutations that confer antibiotic resistance may reduce the fitness of the bacteria in the absence of antibiotics, allowing resistant mutants to be outcompeted by antibiotic-susceptible bacteria. Further research on these approaches can be expected to provide clear principles on the most effective use of β-lactams for treatment while minimising the emergence of resistant bacteria. More broadly, there is limited research on the circumstances in which *P. aeruginosa* acquires β-lactamase genes by HGT. However, as in the clinic, the presence of antibiotics in the general environment will promote antibiotic resistance. Minimising the discharge of β-lactams from clinical settings such as hospitals into wastewater will reduce the selection of antibiotic-resistant bacteria.

An alternative approach to circumventing resistance would be to develop new β-lactams that are unaffected by resistance mechanisms such as β-lactamases, along with new β-lactamase inhibitors. Examples of new β-lactams in development include the carbapenem benapenem [312,313] and a monobactam BOS-228 [312,314]. Combinations of β-lactams with a number of new β-lactamase inhibitors are undergoing clinical trials [315,316]. Diazabicyclooctane-derived β-lactamase inhibitors can inhibit class A and C β-lactamases but have limited activity against class D β-lactamases, whereas enmetazobactam and LN-1-255 show activity against class A, C, and D β-lactamases [316]. Three boronic acid-derived β-lactamases inhibitors VNRX-5236, taniborbactam, and xeruborbactam are also undergoing clinical trials [316]. Of particular significance, taniborbactam and xeruborbactam are pan-β-lactamase inhibitors that can inhibit all classes of β-lactamases, including metallo-β-lactamases which are not targeted by other inhibitors. A novel metallo-β-lactamase inhibitor ANT431 is also at a pre-clinical stage [317,318]. A number of other compounds are also under development, although how many will find their way into the clinic, and whether they will be subject to the same resistance mechanisms as existing compounds, is not yet clear [319,320].

Rational development of new anti-Pseudomonas antibiotics is greatly enhanced by an understanding of the mechanisms of action of, and resistance to, existing antibiotics [320]. Carbapenems can enter *P. aeruginosa* via OprD, but other antibiotics are thought to diffuse through the lipid bilayer of the outer membrane. A better understanding of how β-lactams access the periplasm may enable more effective uptake. One approach that has been explored is to conjugate β-lactams to siderophores, low molecular weight compounds imported into the periplasm by *P. aeruginosa* to enable the acquisition of iron a key nutrient [321,322]. A newly approved cephalosporin-siderophore, cefiderocol (FDA 2019 and European Union 2020), is showing promising results against *P. aeruginosa* [323,324] shown to be active against greater than 95% of *P. aeruginosa* isolates [325] Resistance towards cefiderocol can arise through mutations in AmpC and in TonB-dependent transporters required for entry of the antibiotic into bacterial cells [237,326,327] and the extent to which resistance becomes a clinical problem remains to be determined. Encapsulating β-lactams into liposomes or loading them into nanoparticles for targeted antibiotic release may also increase the local concentration of antibiotics, enhancing entry into the bacterial cells [315,328,329].

As well as β-lactamases, other *P. aeruginosa* proteins contribute to resistance and are potential targets for antibiotics. For example, inhibitors of AmpR would be expected to increase the susceptibility of *P. aeruginosa* to AmpC-susceptible β-lactams, and inhibition of the MexABOprM efflux pump would also increase susceptibility. Phe-Arg-β-naphthylamide (PAβN) is a broad-spectrum efflux inhibitor that reduces MICs towards antibiotics in vitro [330,331]. PAβN is toxic to humans but its effectiveness in vitro demonstrates the potential of efflux pump inhibitors as targets for anti-Pseudomonal therapy.

In conclusion, β-lactams will be key members of the anti-*Pseudomonas* armamentarium for many years to come. Future research on their modes of action and or how to overcome the threat of resistant bacteria will be essential to maintain and maximise their usage.

## Figures and Tables

**Figure 1 pathogens-10-01638-f001:**
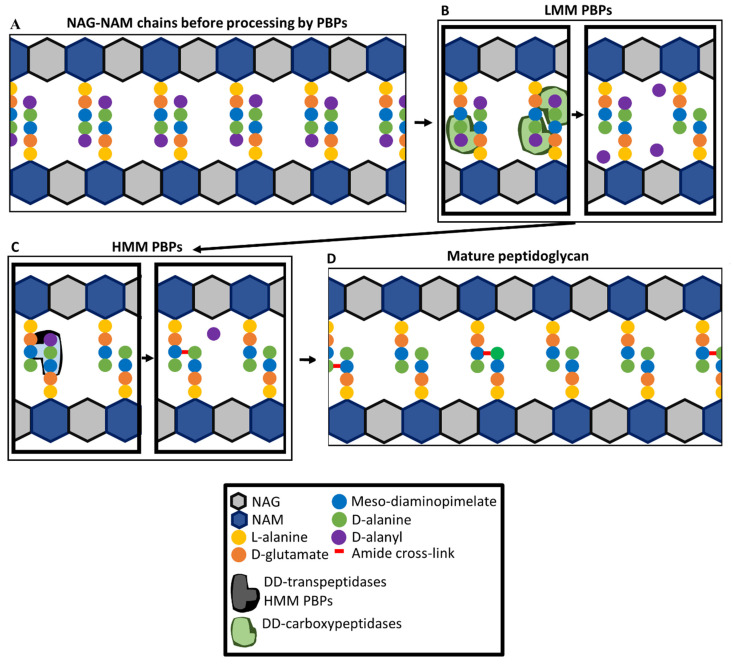
Roles of penicillin-binding proteins (PBPs) in peptidoglycan synthesis. (**A**) NAG-NAM chains are not cross-linked before processing by PBPs. (**B**) LMM PBPs 4, 5 and 7 (DD-carboxypeptidases) cleave terminal D-alanyl residues from some pentapeptides, regulating levels of cross-linking. (**C**) HMM PBP 1a, 1b, 2, 3, 3a (DD-transpeptidases) cross-link pentapeptide-containing side chains to penta-, tetra-, or tri-peptides of adjacent NAG-NAM chains while simultaneously removing terminal D-alanyl residues. (**D**) Mature peptidoglycan contains a mixture of cross-linked and unlinked peptides. NAG, N-acetyl glucosamine; NAM, N-acetyl muramic acid.

**Figure 2 pathogens-10-01638-f002:**
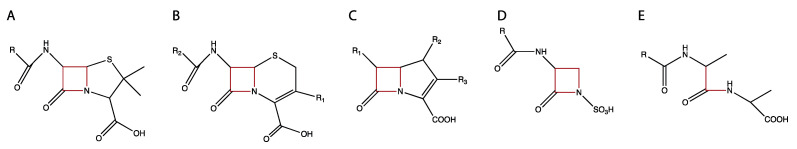
Core structures of β-lactam subclasses used in *P. aeruginosa* treatment, and the terminal D-alanine-D-alanyl residues of peptidoglycan pentapeptide. (**A**) Penicillins. (**B**) Cephalosporins. (**C**) Carbapenems. (**D**) Monobactams. (**E**) D-alanine-D-alanyl residues. The β-lactam ring is indicated in red and mimics the terminal D-alanine-D-alanyl of the peptidoglycan pentapeptide precursor. Figure adapted from [101,107].

**Figure 3 pathogens-10-01638-f003:**
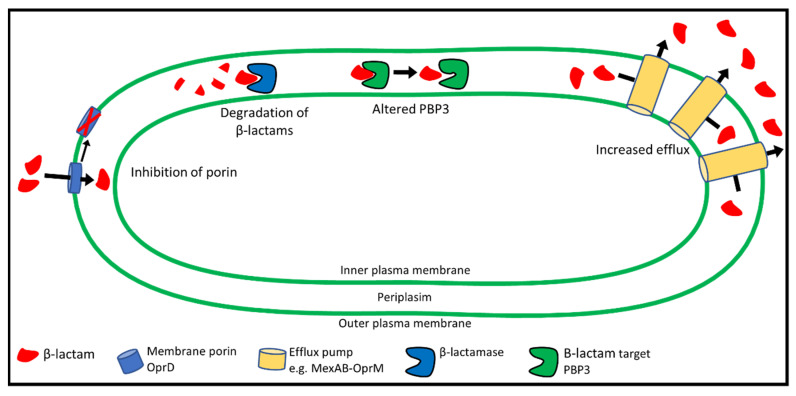
Mechanisms of β-lactam resistance in *P. aeruginosa*.

**Figure 4 pathogens-10-01638-f004:**
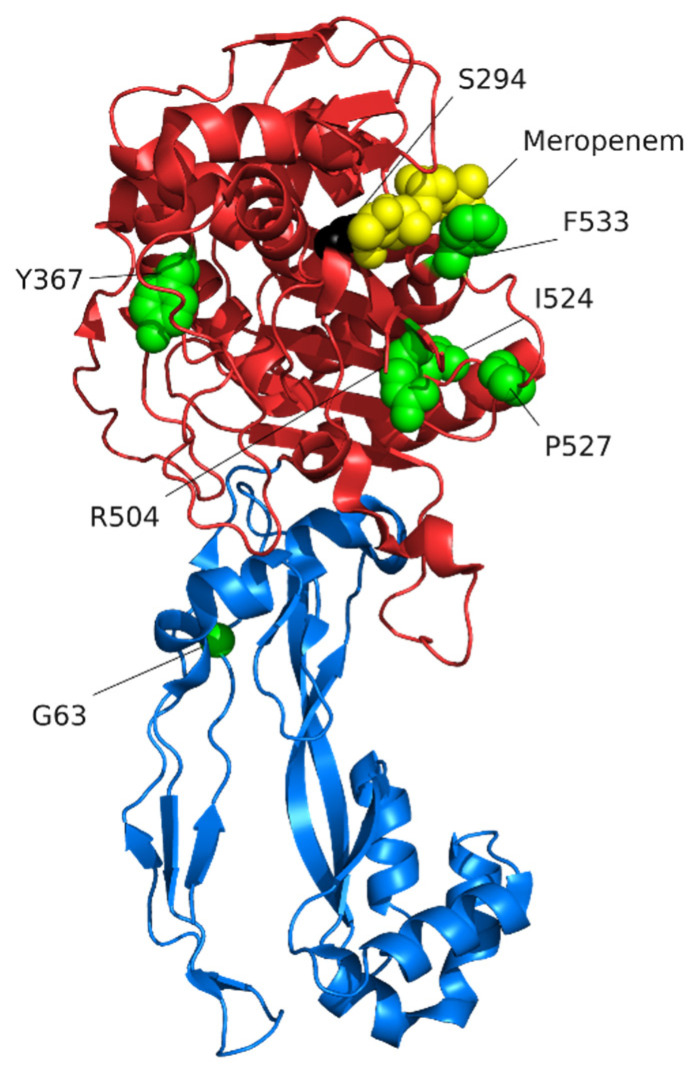
Structure of PBP3 in complex with meropenem. The transpeptidase domain is shown in red. The domain shown in blue is thought to play a role in protein-protein interactions. The membrane-spanning helix and small cytoplasmic part of PBP3 are not included in the structure. Meropenem shown in yellow is bound to the catalytic serine S294 (in black). Amino acid residues that are commonly substituted in clinical isolates are coloured green with side chains displayed. The image is based on protein structure PDB 3PBR_1 [102].

**Figure 5 pathogens-10-01638-f005:**
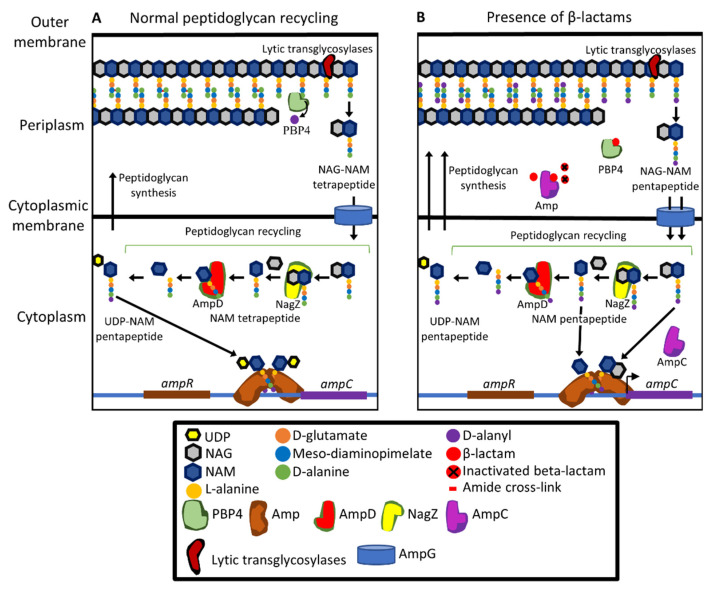
Regulation of *ampC* expression. (**A**) Under normal cellular conditions expression of *ampC* is repressed. LMM PBPs such as PBP4 hydrolyse uncross-linked peptidoglycan pentapeptides to tetrapeptides. During recycling of peptidoglycan, peptidoglycan fragments (the majority being NAG-NAM tetrapeptide cleaved from the NAG-NAM chains by lytic transglycosylases) are imported into the cytoplasm. NAG is removed by NagZ, after which NAM is cleaved from the peptide side chain by AmpD. NAG, NAM and the peptide side chains are used in synthesis of new peptidoglycan. Excess peptidoglycan precursor UDP-NAM pentapeptide formed through recycling as well as *de novo* synthesis binds to the AmpR regulator protein, which acts as a repressor inhibiting *ampC* expression. (**B**) β-lactams cause upregulation of *ampC*. Increased peptidoglycan recycling occurs because of the presence of β-lactams, which also inhibit conversion of tetrapeptides to pentapeptides by LMMs PBPs. The resulting peptidoglycan fragments (primarily NAG-NAM pentapeptide but also NAG-NAM tripeptide [not shown]) are imported into the cytoplasm. In the recycling pathway, AmpD becomes saturated because of increased amounts of peptidoglycan fragments, increasing the intracellular concentrations of the AmpR activator molecules NAG-NAM pentapeptide, NAM-pentapeptide and NAG-NAM tripeptide. Increased export of UDP-NAM pentapeptide for peptidoglycan synthesis also occurs. The activator molecules outcompete UDP-NAM pentapeptide for binding to AmpR and the AmpR-activator complexes trigger increased expression of *ampC*. UDP, uridine diphosphate; NAG, N-acetyl glucosamine; NAM, N-acetyl muramic acid; pentapeptide, L-alanine-γ-D-Glutamate-meso-DAP-D-Ala-D-Ala.

**Figure 6 pathogens-10-01638-f006:**
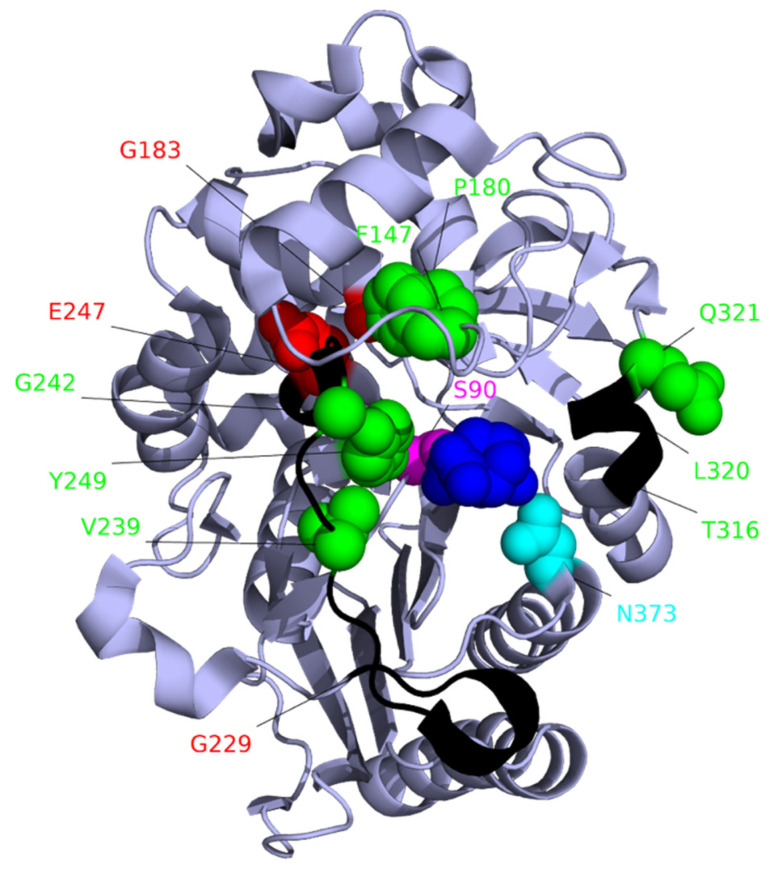
Locations of amino acid variants in AmpC that contribute to β-lactam resistance.

**Table 1 pathogens-10-01638-t001:** Key antibiotics used in the treatment of *P. aeruginosa* [2,42,43,44,45,46,47,48,49,50,51,52,53].

Antibiotic Class	Antibiotic Subclass	Antibiotic	Antibiotic Use
β-lactams ^a^	Penicillins	Ticarcillin Piperacillin	Parenteral or intravenous for treatment of infections
	Monobactams	Aztreonam	Inhalation for long-term treatment of chronic lung infections and intramuscular injection for the treatment of acute infections
	Carbapenems	Imipenem ^b^ meropenem	Intravenous for treatment of acute or chronic infections
	Cephalosporins	Ceftazidime Cefepime Ceftolozone	Inhalation or intravenous for treatment of acute or chronic infections
Aminoglycosides	4,6-di-substituted deoxystreptamine ring	Tobramycin Gentamicin Amikacin	Inhalation or intravenous for treatment of acute or chronic infections
Quinolones	Fluoroquinolones	Ciprofloxacin Levofloxacin	Oral or intravenous intake for treatment of acute infections
Lipopeptides	Polymyxins	Colistin	Inhalation for treatment of chronic lung infections

^a^ β-lactams are often paired with a β-lactamase inhibitor. ^b^ Imipenem is generally administered with cilastatin (inhibitor of DHP-1 an enzyme that metabolises imipenem).

**Table 2 pathogens-10-01638-t002:** Binding affinities of β-lactams for PBPs of *P. aeruginosa*.

	Antibiotic ^a^											
	DOR	IMI		MER	CEF			AZT		CEN	CEP	FAR
PBP	Binding Affinity IC50 (μg/mL) ^b^											
1a	0.5	0.1	0.5	0.5	0.2	0.2	0.8	2	2	19	35	0.23
1b	0.6	0.2	0.5	0.5	3	5	6	2	2	2	0.7	0.15
2	0.06	0.1	0.1	0.05	>32	>32	25	16	16	>250	ND	0.19
3	0.07	0.09	0.1	0.08	0.1	0.1	0.1	0.03	0.03	0.3	41	0.20
4	0.008	0.008	0.01	0.008	2	2	ND	16	16	ND	ND	0.13
5	8	2	2	16	>32	>32	ND	>16	>16	ND	ND	>1
MIC (μg/mL) for *P. aeruginosa* ^c^												
	0.25	1	1	0.5	1	1	0.5	4	4	0.03	16	512
Reference	[110]	[110]	[111]	[110]	[110]	[111]	[112]	[110]	[111]	[112]	[112]	[109]

^a^ DOR, doripenem; IMI, imipenem; MER, meropenem; CEF, ceftazidime; AZT, aztreonam; CEN, Cefsulodin; CEP, Cephalexin; FAR, Faropenem; ^b^ IC50: Half maximal inhibitory concentration, determined through competition assays with the fluorescent β-lactam bocillin FL. ND, not determined. ^c^ MIC: Minimum inhibitory concentration (μg/mL).

**Table 3 pathogens-10-01638-t003:** Antibiotic resistance associated with increased expression of efflux pumps.

Efflux System	Antibiotics Affected by Increased Expression
MexAB-OprM	Aztreonam, other β-lactams ^a^, quinolones, tetracyclines, macrolides, novobiocin and chloramphenicol [172,173]
MexXY-OprM	Aminoglycosides, tetracyclines, β-lactams ^b^ and macrolides [171,172,174,180,181]
MexCD-OprJ	β-lactams ^c^ and fluoroquinolones [170,174]
MexEF-OprN	Imipenem ^d^ and fluoroquinolones [172,177]

^a^ MexAB-OprM exports all β-lactams except imipenem [174,182]. ^b^ MexXY-OprM exports all β-lactams except carbenicillin, sulbenicillin, cefsulodin, ceftazidime, moxalactam, flomoxef, aztreonam, and imipenem and has low substrate specificity for other carbapenems [174]. ^c^ MexCD-OprJ exports all β-lactams except carbenicillin, sulbenicillin, ceftazidime, moxalactam, aztreonam, and imipenem and has low substrate specificity for other carbapenems [174]. ^d^ Increased expression of *mexEFoprN* reduces *oprD* expression [183,184].

## Data Availability

Not applicable as no data was generated for this review.

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
