# Peer review of "β-lactam Resistance in Pseudomonas aeruginosa: Current Status, Future Prospects"

_pathogens, 2021, doi:10.3390/pathogens10121638_

Round 1
Reviewer 1 Report
This review is comprehensive, however, I just have two minor concerns.
- Table 1 needs corrections. Some information withint this table is not enough or wrong.
- Please add more discussions about the novel antibiotics and the antibiotic-resistant P. aeruginosa.
Reviewer 2 Report
(1) In line 123, the use of "...infections can be directly infected with..." is confusing. Please make this sentence clearer.
(2) Stand alone single digits like 2 in line 126 should be spelled out.
(3) Suggestion for the first paragraph in the "Mechanisms of beta-lactam resistance" section: include that you are going to cite examples of all of the general mechanisms that you describe here.
(4) In line 228, you are using "arise through" in two successive sentences; please paraphrase if you can to avoid this.
(5) For the paragraph beginning with line 281, have any of the studies cited described magnitude in reduction of sensitivity? If so, that would be interesting to include here.
(6) In line 291, change "toward" to "to."
(7) Line 312: Is this sentence intended to mean the species altogether, or instead common in every P. aeruginosa genome? I would make this clearer somehow.
(8) Line 346: comma after PA3719.
(9) Line 368 section header: consider replacing colon with "by."
(10) Line 403: should "de novo" be italicized?
(11) Line 418: missing hyphen in beta-lactam.
(12) Line 445: in last sentence, should end it with Author et al., year or is it okay with just the footnote?
(13) Line 497: consider replacing "self-conjugative" with "self-mobilizing."
(14) Paragraph starting with line 533: is there any indication that these mutations in LPS reduce fitness or virulence? That is an important consideration for clinical relevance.
(15) Paragraph starting with line 579: non-antibiotic antimicrobials should also be considered as a way to address the multidrug resistance problem, but this option is absent from the conclusion section and this manuscript. Development of such antimicrobials may be further in the future but still an important area of development.
(16) Paragraph starting with line 604: there is a broader possibility of using non-antibiotic antimicrobials in combination with extant antibiotic therapies, not solely in finding inhibitors of resistance factors such as beta-lactamases. This is potentially a fruitful area of research and development because these "alternative" antimicrobials may have mechanisms of action that are completely different from stand of care antibiotics including the beta-lactams.
(17) Consider replacing "due to" with other word forms such as "because of."
Sorry to be a bit nit-picky in places, but am trying to make suggestions to improve a solid manuscript.
